# Biomarkers Correlated with Tuberculosis Preventive Treatment Response: A Systematic Review and Meta-Analysis

**DOI:** 10.3390/microorganisms11030743

**Published:** 2023-03-14

**Authors:** Haoran Zhang, Zuyu Sun, Yi Liu, Rongrong Wei, Nanying Che

**Affiliations:** 1Biobank of Beijing Chest Hospital, Capital Medical University, Beijing 100054, China; 2Department of Pathology, Beijing Tuberculosis and Thoracic Tumor Research Institute, Beijing Chest Hospital, Capital Medical University, Beijing 100054, China

**Keywords:** tuberculosis, preventive treatment, biomarkers, monitoring, meta-analysis

## Abstract

Background: There is a need to identify alternative biomarkers to predict tuberculosis (TB) preventive treatment response because observing the incidence decline renders a long follow-up period. Methods: We searched PubMed, Embase and Web of Science up to 9 February 2023. The biomarker levels during preventive treatment were quantitatively summarized by means of meta-analysis using the random-effect model. Results: Eleven eligible studies, published during 2006–2022, were included in the meta-analysis, with frequently heterogeneous results. Twenty-six biomarkers or testing methods were identified regarding TB preventive treatment monitoring. The summarized standard mean differences of interferon-γ (INF-γ) were −1.44 (95% CI: −1.85, −1.03) among those who completed preventive treatment (τ^2^ = 0.21; I^2^ = 95.2%, *p* < 0.001) and −0.49 (95% CI: −1.05, 0.06) for those without preventive treatment (τ^2^ = 0.13; I^2^ = 82.0%, *p* < 0.001), respectively. Subgroup analysis showed that the INF-γ level after treatment decreased significantly from baseline among studies with high TB burden (−0.98, 95% CI: −1.21, −0.75) and among those with a history of Bacillus Calmette–Guérin vaccination (−0.87, 95% CI: −1.10, −0.63). Conclusions: Our results suggested that decreased INF-γ was observed among those who completed preventive treatment but not in those without preventive treatment. Further studies are warranted to explore its value in preventive treatment monitoring due to limited available data and extensive between-study heterogeneity.

## 1. Introduction

The World Health Organization (WHO) recommended tuberculosis (TB) preventive treatment as a key intervention strategy [1]. However, in the absence of an effective vaccine, TB preventive treatment is currently a difficulty in global TB control. The implementation of preventive treatment is beset with numerous obstacles due to its lack of objective treatment indication and effect evaluation index, which seriously affects the realization of the global END TB strategy. According to the 2022 WHO TB report, the global number of people provided with TB preventive treatment in 2021 was 3.5 million [2]. In addition, the population of latent tuberculosis infection (LTBI) in China is hundreds of millions [3]. In order to control the TB epidemic in China, developing suitable strategies for LTBI management is of utmost importance. In addition to preventing infection, the scale-up of preventive treatment coverage for the high-risk groups and evaluation of the therapeutic efficacy of LTBI must be addressed [4,5]. The evaluation of efficacy is essential to ensure that the vast number of people initiated on TB preventive treatment can benefit more. The efficacy of currently available preventive treatment ranges from 60 to 90% [6]. Direct observation of the decline in incidence is the standard method used to evaluate the protective effect of the preventive treatment. However, this usually requires a long follow-up period and huge resource costs. From the viewpoint of clinical and public health needs, instant and sensitive biomarkers or tests for monitoring the performance and predicting the outcomes of TB preventive treatment are warranted.

For adult pulmonary TB patients, the WHO recommends sputum smear microscopy and/or culture conversion at the end of the intensive phase of treatment as methods of monitoring treatment response [2,7,8]. Although sputum testing also could be used in evaluating successful TB preventive treatment, there is no instant and sensitive advantage in practice [9]. Additionally, no current WHO guidelines are recommended for the assessment of TB preventive treatment efficacy. A systematic review summarized the available evidence of active TB treatment monitoring biomarkers [10], but no systematic review has evaluated changes in biomarker levels with respect to TB preventive treatment. The requirement for biomarkers in the evaluation of TB preventive treatment stems from the critical aspect: the long and variable natural history of Mycobacterium tuberculosis (MTB) [11,12,13]. Timely efficacy evaluation will guarantee the optimization of TB control strategies and technical pathways to some extent. The main purpose of conducting a centered biomarker investigation is to concisely assess the effectiveness of preventive treatment and thereafter to prevent the development of active diseases [14]. In addition, new drugs, vaccines and other therapies will be required to realize the goal of END TB worldwide [4]. The important role of biomarkers also helps to accelerate intervention development by offering surrogate endpoints. Some studies were conducted to explore the potential biomarkers or assays for predicting clinical outcomes [9,15,16,17,18,19]. The types of specimens (e.g., blood and urine), the methods of testing (e.g., ELISA and transcriptomic sequencing) and the targeted biomarkers (e.g., gene expression signatures and cytokines) were heterogeneous [20,21]. Effective supporting evidence needs to overcome the study heterogeneity, especially by appropriately differentiating between clinical heterogeneity such as preventive treatment regimen [15,16,22,23]. No previous systematic review has addressed this. However, knowledge about specific biomarkers or assays that might represent promising options to optimize preventive treatment monitoring is still largely limited. The most valuable biomarkers are given great expectations of being directly involved in pathogenesis or protection and for which changes early during the preventive treatment stage can be related to the pharmacology of the intervention. This dynamic of the response and its relation to short- and long-term outcomes should be further evaluated by means of treatment trials. In this sense, a systematic summary of the related evidence on biomarkers during TB preventive treatment will benefit the constitution of efficacy indicators. More importantly, candidate biomarkers need to verify their influence on the delivery of routine care. Therefore, to better guide the recommendations for the use of biomarkers for TB preventive treatment response, this systematic review aims to summarize the evidence for biomarkers that is related to TB preventive treatment by means of systematic review and meta-analysis.

## 2. Materials and Methods

### 2.1. Methods

The updated Preferred Reporting Items for Systematic Reviews and Meta-analyses (PRISMA 2020, 27-item checklist) guideline was used in reporting our findings [24] (Appendix A). The study was registered in PROSPERO (CRD42023393104). No patients or members of the public were directly involved in this research study.

### 2.2. Search Strategy

Electronic databases PubMed, Web of Science and the Embase were searched to obtain articles addressing the biomarkers that correlated with TB preventive treatment response from database inception to 9 February 2023. The searches included combinations of key blocks of terms involving medical subject heading terms and text words: “tuberculosis infection”, “latent tuberculosis infection”, “LTBI”, “latent tuberculosis” and “latent TB” to represent the exposed population; “biomarkers”, “biomarker”, “markers” and “marker” to indicate the outcomes index; and combined them with terms related to preventive treatment, such as “prophylaxis”, “prevention”, “prophylactic treatment” or “prophylactic therapy”. The search strategies are detailed in Appendix A. The references list of relevant systematic reviews and eligible studies was manually examined to identify additional literature.

### 2.3. Eligibility Criteria

The criteria for inclusion in the meta-analyses were as follows: original articles investigating the longitudinal changes in biomarker levels during TB preventive treatment irrespective of study design; participants should initially test MTB-positive and then receive a monitored course of TB infection preventive treatment; and level of biomarkers should be measured at two or more time points. The decision to include only articles published in English was due to language capabilities of study team. If the study was reported in duplication, the version first published or which provided more detailed information was included. Review articles, animal studies, case reports, commentaries/editorials, mathematical modeling studies, conference abstracts and studies addressing active pulmonary TB treatment or exploring diagnostic biomarkers were excluded. We also excluded studies in which required data was unavailable, there was no full text, or there was no response to the request for data from authors. To minimize the potential bias caused by too small a sample size, articles with a sample size of fewer than 10 participants were excluded from the meta-analyses. In addition, to improve validity of data, non-peer-reviewed articles in preprint databases were excluded.

### 2.4. Study Selection and Data Extraction

All publications identified from the search strategy were imported into the reference management database EndNote (version X9, Clarivate^TM^, Philadelphia, PA, USA). After duplications were removed, studies were screened in two stages: first by title and abstract and then by full-text article. Two researchers (ZHR and SZY) independently screened each title, abstract, full text and data extraction, with discrepancies resolved by consensus with a third researcher (LY). All full texts were checked against eligibility criteria (WRR).

During the extraction process, a predetermined proforma in Microsoft Excel Version 16.54 was used. All key extracted data were reviewed and quality-checked at the end of the data extraction phase by the same two researchers. For the quantitative assessment, we only extracted data on biomarkers when their quantitative level changes were reported by five or more studies. The levels of biomarkers and measures of spread data were extracted directly from the texts or tables when available. If not available, the data were extracted directly from available figures. In addition, for each included study, extracted data on study characteristics comprised of study site, study design, first author and published year, and study population. Participant data comprised age, sex, history of prior TB disease and anti-TB treatment, Bacillus Calmette–Guérin (BCG) vaccination and HIV-infected or acquired immune deficiency syndrome. Preventive treatment-related data included regimen of preventive treatment, number of participants receiving preventive treatment with serial test results and intervals between follow-ups. Outcome-related data comprised assay type, method and numbers of measurement, biomarkers levels of baseline and follow-up and follow-up period.

### 2.5. Assessment of Quality and Risk of Bias

Risk of bias was assessed using the Quality Assessment of Diagnostic Accuracy Score 2 (QUADAS-2). This tool consists of four domains: patient selection, index test, reference standard, and flow and timing. Each domain was evaluated using a set of guiding questions (Appendix A). Items were scored as “high concern,” “low concern,” or “unclear concern.” The overall risk of bias was evaluated as “high risk” for studies with more than one area of high concern, “low risk” for those with two or more areas of low concern and no high risk and “unclear risk” for those with three or more areas of unclear concern and no high risk. Two reviewers (LY and WRR) appraised the risk of bias in the results of all studies that met inclusion criteria independently, with discrepancies resolved by discussion with a third reviewer (ZHR).

### 2.6. Outcomes

The prespecified outcome in the meta-analysis was association of dynamic changes in biomarkers levels with respect to TB preventive treatment. In addition to declining the incidence of active TB disease, infection clearance could be used for estimating for efficacy of preventive treatment. However, due to lack of golden standard for defining MTB infection, current infection testing could not absolutely reflect the real infection status. Therefore, we only indirectly evaluated the response to TB preventive treatment by comparison across groups. For instance, we compared the biomarker levels in different time points between different subgroups: preventive treatment group and untreated controls; infection testing positive group, negative groups or healthy control group; and comparison of baseline and post-treatment levels in the same population group.

Biomarker outcomes were required to collect more than one time point, at least with results at baseline and after preventive treatment. If more than two time points were reported, we extracted baseline results and the last testing results after preventive treatment. If more than two biomarkers in one study were reported, data for each study would be extracted. If two or more measurement tools were used in one study, corresponding results would be extracted and described, respectively.

### 2.7. Data Synthesis and Analysis

The extracted data were first transformed and standardized as mean and standard deviation (SD) values. The Box–Cox (BC) method [17,24] was applied to estimate the sample mean and SD from studies that reported the median accompanied by first and third quartiles. Standardized mean difference (SMD) was calculated as the effect size for each biomarker and summarized it using the random-effects model based on the Hartung–Knapp–Sidik–Jonkman method after adjustment to Hedges’ g [25,26]. SMD was applied because different testing methods would be used to measure the same outcomes. Confidence intervals (CIs) were converted to SDs. When measures of variation were missing for mean differences within each treatment arm of a given study but a test of difference between treatment arms was reported, we converted F-statistics, t-statistics and *p*-values to standard errors and SDs. Here, t was taken as the square root of F, and it was assumed that the SDs of the mean differences in each treatment arm were equal. However, if the study showed no test of difference between groups, we used the highest SD recorded in the same meta-analysis for each treatment arm instead of the study’s own data. At least two studies were required for each meta-analysis [27]. When studies included more than two intervention groups, we excluded irrelevant groups or combined relevant groups as recommended in the *Cochrane Handbook* in order to avoid arbitrary decisions [28]. For example, a regimen of isoniazid (INH) and rifapentine for 3 months (3HP) was compared with a regimen of INH for 6 months (6H) and 9H regimen group. Using the formula provided for combining two treatment groups (*Cochrane Handbook* 6.5.2.10, 23.3.4), when we aimed to target 3HP regimen as a subgroup, we calculated the combined mean difference and SD for 6H regimen and 9H regimen groups as a single control group.

As different biomarkers or assays were included, we performed statistical pooling restricted to biomarkers or assays, which had two or more studies that quantitatively presented the data of measures at different follow-up time points. We investigated heterogeneity by performing subgroup analyses on the following variables: TB burden of country of enrollment, age of participants (years), sex, history of BCG vaccination and sample size. Heterogeneity was assessed using the Cochrane Q test and quantified as I^2^ values and τ^2^ (the restricted maximum likelihood estimator was used to estimate this between-study variance) [10,29]. Heterogeneity was considered significant if the *p*-value of Cochran’s Q test was <0.10 or if the I^2^ statistic was ≥50% [30]. Publication bias for one specific outcome rather than studies was shown using a funnel plot. Small-study effect was assessed by calculating Egger’s test score and Begg’s rank correlation analysis [31,32]. Meta-analyses were carried out using STATA Meta-Analysis (V2.0, Biostat, Englewood, NJ, USA). A two-sided *p* < 0.05 was considered to be statistically significant.

## 3. Results

### 3.1. Study Identification and Selection

As shown in Figure 1, a total of 5718 articles were obtained by database searches using different combinations of key terms. After removing duplicates, 5325 records were screened by title and abstract, of which 121 full texts were retrieved for detailed evaluation. Appendix A shows a list of excluded studies with reasons. Of the excluded 105 articles which underwent full-text screening, 59 were excluded due to no information on TB preventive treatment, 18 due to mathematics modeling studies, 9 due to no serial data on TB preventive treatment monitoring, 5 due to no biomarkers being tested, 10 due to no data on treatment monitoring, 3 due to the required data being unavailable and 1 due to a sample < 10. Finally, 11 studies were eligible for data extraction and quantitative analysis [33,34,35,36,37,38,39,40,41,42,43] and are listed in Appendix A. 

### 3.2. Quality and Risk of Bias Assessment

The QUADAS-2 assessments are summarized in Appendix A and Figure 2. When considering the four main categories of the QUADAS-2 tool, only two studies showed an overall low risk of bias. Specifically, the risk of bias for patient selection was high for studies that used a case-control study design (*n* = 4). Regarding preventive treatment monitoring reference standards, the results which were interpreted without knowledge of the reference standard received a “high risk of bias” (*n* = 3). Most studies did not report whether the reference standard was blinded while interpreting the results of the index (*n* = 6). Finally, the “flow and timing” of the study were generally at a “low risk of bias” as most samples for testing were either processed immediately or frozen.

### 3.3. Characteristics of Studies Included in the Study

Table 1 summarizes the characteristics of the included studies. The publication of the studies occurred between 2006 and 2022, and the sample size varied from 11 to 2618. Six studies were conducted in Asia [33,34,35,36,37,38], three in Europe and two in Africa [39,40,41,42,43]. Six studies were from high TB burden countries [33,34,37,38,39,40], two from middle TB burden countries [35,36] and three from low TB burden countries [41,42,43]. Five studies were of a randomized controlled trial (RCT) design [33,34,37,38,40], three were prospective studies [35,39,42], two were case-control studies [41,43], and one was a cross-sectional study [36]. Four studies were conducted in close-contact subjects with TB patients [36,39,40,42], six in healthy adults [33,34,35,37,38,43] and one in both adults and students [41]. Two studies indicated study participants had a prior history of TB [37,43]. Seven studies showed participants had previously received the BCG [33,36,39,40,41,42,43]. No studies included participants living with HIV.

Across all included studies, different biomarkers or testing methods were identified (*n* = 26). Of all the biomarkers, interferon-γ (IFN-γ) was the most frequently analyzed biomarker for TB preventive treatment monitoring [33,34,36,40,43] (*n* = 5). Studies that examined well-established diagnostics, such as QuantiFERON-TB Gold Plus (QFT-Plus), were not excluded as no previous systematic reviews have characterized their capabilities in TB preventive treatment monitoring.

### 3.4. Biomarker Levels before and after TB Preventive Treatment

For biomarkers where there were two or more studies that numerically presented the dynamic levels at different follow-up time points, then meta-analysis was further performed. For INF-γ, the summarized SMD was found to be −1.44 (95% CI: −1.85, −1.03) for those who completed preventive treatment (τ^2^ = 0.21; I^2^ = 95.2%, *p* < 0.001) and −0.49 (95% CI: −1.05, 0.06) for those without preventive treatment (τ^2^ = 0.13; I^2^ = 82.0%, *p* < 0.001) (Figure 3A). With respect to QFT-Plus testing, which reflected the levels of TB1 and TB2 tube antigens, we summarized their status after treatment, such as reversion or conversion of MTB infection, based on the reported results of included studies. The summarized rate of reversion was observed to be 9.3% (4.3–14.3%) after preventive treatment (τ^2^ = 0; I^2^ = 0%, *p* = 0.613) (Figure 3B). In addition, four cytokine proteins, IL-2, IL-5, IL-13 and IL-17a, were analyzed using quantitative synthesis (Figure 3C). The results of this meta-analysis found that the CI for these four biomarkers crossed the null and therefore did not reach statistical significance.

As shown in Table 2, despite the heterogeneity between the included studies, the summarized biomarker levels before and after TB preventive treatment were found to be significantly influenced by the TB burden of the country of enrollment, age of participants, male gender, history of BCG vaccination and sample size (*p* < 0.001) in the stratified analyses. In addition to the age of participants, male gender and sample size, the changed level was found to be particularly significant in the studies with a high TB burden (−0.98; 95% CI: −1.21, −0.75) (τ^2^ = 0.04; I^2^ = 92.1%, *p* < 0.001) and among participants with BCG vaccination history (−0.87; 95% CI: −1.10, −0.63) (τ^2^ = 0.06; I^2^ = 92.1%, *p* < 0.001). 

No evident publication bias was observed when all 11 studies were evaluated using Begg’s rank correlation analysis (*p* = 0.216) and Egger’s weighted regression analysis (*p* = 0.389), but the funnel plot seemed to be asymmetric (Appendix A). 

## 4. Discussion

To our knowledge, this is the first meta-analysis aiming to evaluate biomarkers that correlated with TB preventive treatment response. The findings of the included studies were found to be frequently heterogeneous. INF-γ, QFT-Plus and cytokines, including IL-2, IL-5, IL-13 and IL-17a, were commonly used for evaluating responses to TB preventive treatment. The summarized changes in INF-γ levels with respect to TB preventive treatment were found to be −1.44 (95% CI: −1.85, −1.03) for those who completed preventive treatment (I^2^ = 95.2%, *p* < 0.001). In addition, the summarized reversion rate of QFT-Plus was observed to be 10.0% (1.0–19.0%) after preventive treatment (I^2^ = 0%, *p* = 0.613). However, due to the limited included studies, whether the identified biomarkers or testing method in our study could be used to monitor TB preventive treatment efficacy in practice should be further explored.

Globally, the treatment monitoring of LTBI is an essential part of TB control. In recent years, increasing numbers of trial studies have reported the efficacy of different TB preventive treatment regimens across various populations [44,45,46,47]. However, previous meta-analysis has rarely examined the association of biomarkers with TB preventive treatment response. Some systematic reviews only assessed the biomarkers that correlated with active pulmonary TB treatment responses [48]. We chose to focus on the performance of potential biomarkers in preventive treatment because there is currently a lack of biomarkers and assays for clinicians to evaluate the effectiveness and characteristics of preventive treatment. Therefore, to date, the association of accessible biomarkers with preventive treatment response is still equivocal, leaving a large knowledge gap on this topic. In our meta-analysis, we found that the level of INF-γ decreased along with preventive treatment. However, a number of studies, even previous meta-analyses, have shown that dynamic changes in INF-γ levels were not associated with preventive treatment [37,49,50]. Inconsistently, our results indicated that INF-γ levels decreased significantly among participants with preventive treatment (−1.44, 95% CI: −1.85, −1.03) but not among those without preventive treatment (−0.49, 95% CI: −1.05, 0.06). One of the possible explanations is that we directly used the levels of blood INF-γ rather than testing the reversion indicator. A testing method such as QuantiFERON-TB Gold In-Tube (QFT) could be a potential reason because a simple “wobble” around the cut-point due to imperfect test reproducibility could induce misclassification [51,52]. More importantly, the underlying mechanism of TB testing reversion is still unclear [50] and needs further exploration. The declined levels of INF-γ after TB preventive treatment may help with clinical decision-making by identifying people who respond favorably to treatment, though further analyses are needed to characterize how this differs between those who respond to treatment and those who do not respond to treatment or are lost at follow-up. Vice versa, the ability to detect the up- or down-regulation of INF-γ may allow for the simpler and earlier identification of people who respond both favorably and unfavorably to treatment. Although the subgroup analyses showed consistent results to some extent, the number of included studies in these subgroups was low. 

Additionally, QFT-Plus was not excluded from our analysis because its performance has been scarcely evaluated in TB preventive treatment by previous meta-analyses. QFT-Plus, as a new generation of QFT, was developed with two TB-specific antigen tubes (TB1 and TB2). TB1 tube induces a specific CD4+ T-cell response, and TB2 was designed to induce IFN-γ production by both CD4+ and CD8+ T cells. It has been reported that QFT-Plus has increased sensitivity compared to QFT with some capability of identifying recent infection among contacts [53,54]. Therefore, we anticipated a promising performance of QFT-Plus in monitoring the TB preventive treatment efficacy. A previous study involving 6H or rifampicin for 3 months conducted in Italy showed that TB preventive treatment significantly decreased INF-γ levels in response to the antigen present in QFT-Plus tests in LTBI and TB-active patients [43]. Unfortunately, the summarized QFT-Plus results in the present meta-analysis showed a lower rate of reversion (9.0%) compared to previous serial testing results (13.0%) without preventive treatment but were significantly higher than QFT (5.9%). Based on only three studies, we could not obtain more reliable estimates of the associations between QFT-Plus and preventive treatment responses. As for the policy implications, the evidence to guide practitioners and clinicians on the monitoring efficacy of interventions continues to be constrained by limited confirmatory biomarkers. There is no recommendation for TB preventive treatment evaluation in the guidelines from a national or regional level or even from the WHO. 

A lack of good-quality evidence remains a barrier to more conclusive findings. While only two included studies were at low risk of bias, as shown in Appendix A, imprecision may reduce our certainty in the outcomes, further highlighting the need for well-designed trials. Our pooled results were somewhat clinically meaningful but might have been overestimated owing to the small-study effect [55]. The assessment of publication bias was restricted by the limited number of studies for some comparisons; therefore, meta-regression was not performed in our study. Although no evident heterogeneity was observed between the included studies, as indicated by the τ^2^ and I^2^ estimates, potential publication bias was found using the visual inspection of asymmetry in the funnel plot, indicating uncertainty around the observed effect estimates. As suggested by our subgroup analyses, it might be partly explained by the high TB burden of the country. In the present meta-analysis, studies from countries with a high TB burden account for a moderate proportion. In such high-transmission settings, repeated exposure to MTB or re-infection might occur more often. However, we have no clue to speculate on various IFN-γ responses to treatment in populations from different regions. Another concern is that no history of BCG vaccination was associated with declined IFN-γ levels after treatment. This was consistent with previous results that BCG vaccination did not affect QFT performance [56]. In addition, some other factors such as age, sex or sample size might be associated with the heterogeneity. This highlights the need for updating and describing the study design more in detail. 

There are some limitations in this study. First, only articles published in English were identified in our study, thus a potential language bias might exist. Second, we extracted and analyzed the rawest available data in each study where possible, standardized these data using SMD and then performed subgroup analyses to validate the findings. Despite these precautions, some degree of imprecision was still possible in the pooled effect sizes driven by variations in the aggregate data. Accessing individual participant data could considerably improve the precision, which we strongly recommend in future research. Third, because not all necessary information could be obtained from all the studies included, related stratifications (e.g., by prior history of TB) could not be made. Fourth, few included studies and small sample sizes limited data analysis and generation of our results. Each biomarker identified in this study has only been evaluated in a limited number of studies, precluding us from meta-analyzing the dynamic change of these markers throughout treatment. Additional well-conducted studies may be required to identify and validate surrogate biomarkers for the field to advance.

In conclusion, our results show low certainty in supporting the use of INF-γ and QFT-Plus for evaluating TB preventive treatment responses. These preliminary data may help inform future studies to investigate these biomarkers in a more rigorous and standardized manner. In the era of prevention-centered care and to give strong recommendations and better guidance for clinical operations, further studies are warranted to determine the potential applicability of accessible biomarkers to field trials; only in this way can their particular role in monitoring and even diagnosis be served.

## Figures and Tables

**Figure 1 microorganisms-11-00743-f001:**
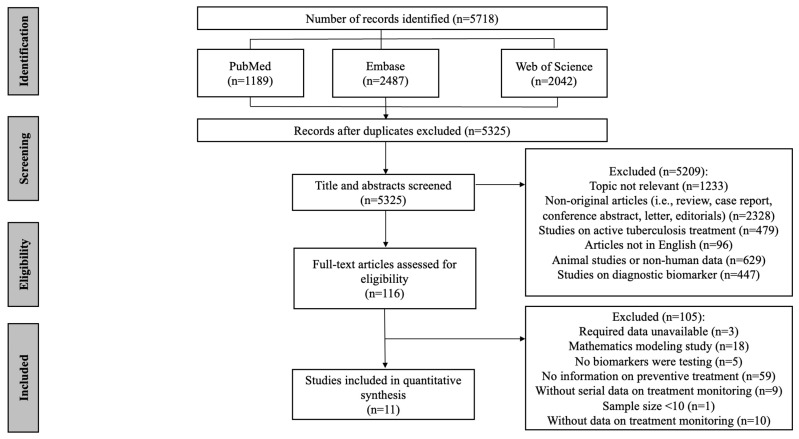
Flow chart of the included study identification.

**Figure 2 microorganisms-11-00743-f002:**
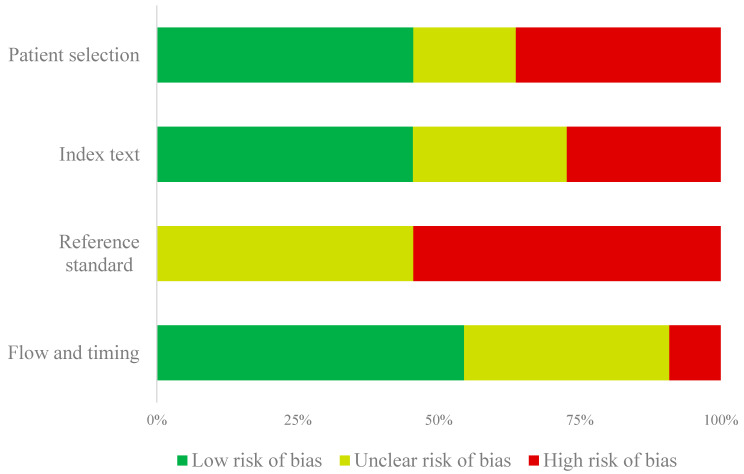
Summary of the QUADAS-2 risk of bias assessment.

**Figure 3 microorganisms-11-00743-f003:**
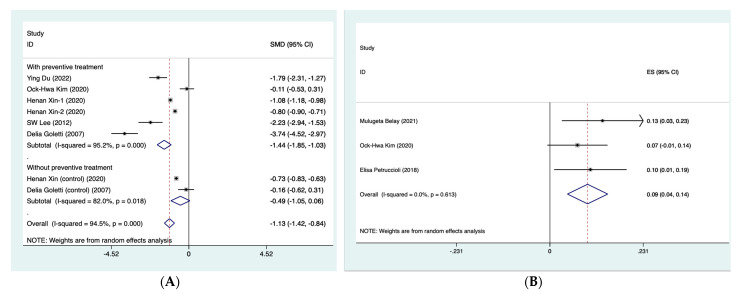
Forest plot of meta-analysis on summarized SMD of INF-γ (**A**) [35,36,37,40,42], summarized rate of reversion QFT-Plus testing (**B**) [33,35,43] and summarized SMD of 4 cytokine proteins (**C**) [34,39]. CI, confidence interval. INF-γ, interferon-γ. QFT-Plus, QuantiFERON-TB Gold Plus. SMD, standard mean difference.

**Table 1 microorganisms-11-00743-t001:** Characteristics of studies included in the meta-analysis.

First Author (Year)	Country (TB Burden)	Study Design	Study Participant	Age (Years)	Male (%)	Sample Size	Prior History of TB (%)	BCG Vaccination (%)	Preventive Treatment Regimen	Assay Name	Outcome at Baseline (Cutoff Value)	Outcome at Follow-Up (Cutoff Value)	Follow-Up Period
Ying Du (2022) [40]	China (high)	RCT	Rural residents	61.8 (mean)	60.0	40	0	-	Rifapentine plus isoniazid 600 mg for 6 weeks	QFT	INF-γ (0.35 IU/mL): 1.26 (0.95–2.39) C4BPA: 108.5 S100A9: 122.5	INF-γ (0.35 IU/mL): 0.06 (0.00–0.13) C4BPA: 74.1 S100A9: 69.8	24 months
Mulugeta Belay (2021) [33]	Ethiopia (high)	Nested prospective study	Household contacts	-	52.0	284	0	40.0	6 months of 300 mg isoniazid and 50 mg pyridoxine daily	Label-free quantitative protein mass spectrometry	M tuberculosis complex, n (%) DNA: 41 (95.0); QFT-Plus: 17 (40.0)	M tuberculosis complex DNA, n (%): 23 (53.0) QFT-PLUS: 13 (30.0)	6 months
Xuefang Cao (2021) [39]	China (high)	RCT	Rural residents	61 (50–69)	68.3	95	0	Treated participants: 53, untreated controls: 28	6 weeks of twice-weekly rifapentine (RPT) plus INH	Digital polymerase chain reaction	IL-1α: 120.8 (−5.7–421.6) IL-8: 12242.9 (1103.1–37141.7) IFN-γ (0.35 IU/mL): 121.9 (−182.5, 1170.2) IL-2: 162.9 (34.2, 886.9) IL-5: 63.5 (27.5, 220.8) IL-13: 4.9 (−0.6, 15.5) IL-17a: 25.4 (4.5, 73.1)	IL-1α: 88.43 (−12.6, 323.2) IL-8: 4604.9 (−618.3, 16283.7) IFN-γ (0.35 IU/mL): −10.7 (−283.9, 777.0) IL-2: 52.7 (4.5, 267.3) IL-5: 65.5 (22.2, 153.4) IL-13: 2.0 (−0.6, 7.3) IL-17a: 27.2 (−7.3, 51.4)	1 week after treatment
Ock-Hwa Kim (2020) [35]	Korea (medium)	Prospective study	Adults	47.6 ± 11.4	34.1	44	0	-	4 months of isoniazid and rifampin or up to 6 months of rifampin	QFT	IFN-γ (0.35 IU/mL): 3.395 QFT-Plus TB1: 3.060 IU/mL (0.35 IU/mL) QFT-Plus TB2: 2.880 IU/mL (0.35 IU/mL) QFT-Plus: 100%	IFN-γ (0.35 IU/mL): 6.804 QFT-Plus TB1: 2.905 IU/mL (0.35 IU/mL) QFT-Plus TB2: 3.880 IU/mL (0.35 IU/mL) QFT-Plus: 93.2	6 months
Henan Xin (2020) [37]	China (high)	RCT	Rural residents	50–70	55.0	Group A: 910, Group B:890, Group C: 818	1.0		Group A: 8 weeks of once-weekly rifapentine (RPT) plus isoniazid (INH); group B: 6 weeks of twice-weekly RPT plus INH; group C: untreated controls	Elisa kit for human cytokine	IFN-γ (0.35 IU/mL): Group A: 1.5 (0.7–3.5) Group B: 1.3 (0.6–3.5) Group C: 1.4 (0.7–3.2)	IFN-γ (0.35 IU/mL): Group A: 0.8 (0.3–2.4) Group B: 0.8 (0.4–2.3) Group C: 0.8 (0.3–2.2)	24 months
Haoran Zhang (2020) [38]	China (high)	RCT	Rural residents	69 (65–73)	50.0	63	-	-	8-week regimen of once-weekly RPT plus INH, 6 weeks of twice-weekly RPT plus INH	QFT	IFN-γ (0.35 IU/mL): 16.5 (11.9–23.0)	IFN-γ (0.35 IU/mL): 13.0 (10.7–14.5)	1 week after treatment
Elisa Petruccioli (2018) [43]	Italy (low)	Case-control	HIV-uninfected patients	38 (24–50)	54.0	46	Microbiologically confirmed: 68.0, clinical diagnosis: 32.0	59.0	INH for 6 months or INH and rifampicin (RIF) for 3 months	QFT-Plus	TB1 or TB2, n (%): TB1 or TB2: 46 (100.0) TB1 and TB2 43 (93.0) Only TB1 1 (.0) Only TB2 2 (4.0) TB1 44 (96.0) TB2 45 (98.0)	TB1 or TB2, n (%): TB1 or TB2: 40 (87.0) TB1 and TB2 37 (80.0) Only TB1 1 (2.0) Only TB2 2 (4.0) TB1 38 (83.0) TB2 39 (85.0)	6 months
Irene Andia Biraro (2015) [34]	Sub-Saharan Africa (high)	RCT	Household contacts	24 (median)	37.0	24	-	IPT group: 17 (71.0), no IPT group: 11 (48.0)	Isoniazid plus pyridoxine daily for 6 months	QFT	IFN-γ (pg/mL): 648 (259.6–2605.4) TNF-α: 6.8 (1–74.6) IL-2: 291.2 (80.6–444.4) IL-5: 1 (1–1) IL-13: 2.1 (1–10.6) IL-10:1 (1–1) IL-17a: 1 (1–1.8) IL-17f 1 (1–1.9) IL-21: 1 (1–1) IL-22: 1 (1–4.9) CFP-10: 2700 (2050–4900) ESAT-6: 3700 (2550–5600)	IFN-γ (pg/mL): 541.1 (290.8, 791.3) IL-2:193.9 (108.5, 279.3) TNF-α: 55.6 (−6.3, 117.6) IL-5:1.1 (0.8, 1.4) IL-13: 5.1 (0.9, 9.4) IL-10: 3.3 (−0.8, 7.6) IL-17a: 1.5 (0.7, 2.3) IL-17f: 10.2 (−6.1, 26.6) IL-21: 123.8 (−133.9, 381.6) IL-22: 6.1 (−0.2, 12.6) CFP-10: 3.4 (3.3, 3.5) ESAT-6: 3.5 (3.4, 3.7)	6 months
SW Lee (2012) [36]	Korea (medium)	Cross-sectional	Close-contact soldiers	21 (20–24)	100.0	26	-	76.9	INH and rifampicin daily for 3 months	QFT	IFN-γ (0.35 IU/mL): 3.6 ± 3.4	IFN-γ (0.35 IU/mL): 0.8 ± 1.1	3 months
Delia Goletti (2007) [42]	Italy (low)	Prospective study	Close contacts	No past exposure: 31, past exposure: 52	33.0	33	-	No INH, no past exposure: 2 (33), past exposure: 1 (20), INH, no past exposure: 11 (46), past exposure: 1 (11)	INH for 6 months	QFT	PHA: 14.7 ± 2.7 PPD: 17.6 ± 2.8 IFN-γ (cutoff value: NA: 17.5 ± 2.7 RD1 proteins: 12.5 ± 2.6 RD1 peptides: 9.2 ± 1.7	PHA: 14.8 ± 2.8 PPD: 14.1 ± 2.5 IFN-γ (cutoff value: NA: 5.2 ± 1.3 RD1 proteins: 2.6 ± 0.9 RD1 peptides: 0.9 ± 0.1	6 months
Katie Ewer (2006) [41]	UK (low)	Case-control	Adults, students	Adults: 47.0 (31–61), students 14.0 (11–15)	Adults: 18.0, students: 58.0	Adults: 11, students: 38	-	Adults: 9 (82.0), students: 30 (79.0)	3-month course of rifampin and isoniazid	Elisa based on RD1 selected peptide and proteins	Adults: RD1 147 (93 to 234), ESAT-6 18 (4 to 96), CFP-10 39 (9 to 171). Students: RD1 247 (173 to 354), ESAT-6 43 (20 to 93), CFP-10 57 (25 to 130)	Adults: 0. Students: RD1 72 (39 to 132), ESAT-6 51 (29 to 89), CFP-10 5 (0 to 74)	18 months

BCG, Bacillus Calmette–Guerin. INH, isoniazid. INF-γ, interferon-γ. QFT, QuantiFERON TB Gold In-Tube. RCT, randomized controlled trial. SMD, standard mean difference. TB, tuberculosis.

**Table 2 microorganisms-11-00743-t002:** Stratified analysis of summarized INF-γ levels before and after TB preventive treatment.

	No. of Studies ^†^	Pooled SMD (95% CI)	Heterogeneity		
			I^2^	*p*	τ^2^
TB burden of country of enrollment ^§^					
High	4	−0.98 (−1.21, −0.75)	92.1	<0.001	0.04
Medium	2	−1.15 (−3.23, 0.93)	96.1	<0.001	2.17
Low	2	−1.94 (−5.45, 1.57)	98.3	<0.001	6.31
Age of participants (years)					
<60	4	−1.53 (−3.06, −0.01)	96.6	<0.001	2.31
≥60	4	−0.98 (−1.21, −0.75)	92.1	<0.001	0.04
Male (%)					
<50	3	−1.13 (−1.42, −0.84)	97.2	<0.001	2.50
≥50	5	−1.10 (−1.35, −0.85)	92.3	<0.001	0.06
History of BCG vaccination					
No	5	−0.87 (−1.10, −0.63)	92.1	<0.001	0.06
Yes	3	−2.03 (−4.18, 0.12)	97.1	<0.001	3.49
Sample size					
<50	5	−1.58 (−2.79, −0.37)	95.9	<0.001	1.82
≥50	3	−0.87 (−1.08, −0.67)	92.3	<0.001	0.03

^§^ Based on 2022 data. ^†^ If the included studies had multiple treatment or control groups, the pooled analysis was undertaken based on each comparison. Therefore, the sum would be not the total of included studies. BCG, Bacillus Calmette–Guerin. CI, confidence interval. SMD, standard mean difference. TB, tuberculosis.

## Data Availability

This study is registered at https://www.crd.york.ac.uk/prospero/#myprospero (accessed on 30 January 2023) with identifier CRD42023393104. The corresponding authors can provide, upon request, data used for analyses and analytic codes and any other materials used in the review after applying necessary measures to guarantee that no individual is identified or identifiable.

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
