# Peer review of "Biomarkers Correlated with Tuberculosis Preventive Treatment Response: A Systematic Review and Meta-Analysis"

_microorganisms, 2023, doi:10.3390/microorganisms11030743_

Round 1

Reviewer 1 Report

The authors searched the three database and meta-analysis of seven publications, indicating that INF-γ is potential biomarker for predicting TB preventive treatment. There are some issues need to be addressed before the publication.

1.       Only 28 references were cited, more references should be cited, especially in Introduction part.

2.       No Figures were found in the Main text.

Author Response

Response to Reviewer 1 Comments

The authors searched the three database and meta-analysis of seven publications, indicating that INF-γ is potential biomarker for predicting TB preventive treatment. There are some issues need to be addressed before the publication.

Point 1: Only 28 references were cited, more references should be cited, especially in Introduction part.

Response 1: Thanks for your meaningful suggestion. We added more 30 references in the main text.

Point 2: No Figures were found in the Main text.

Response 2: We apologized for it, and we added the Figures 1 to 3 and Tables 1 to 2 at the end of manuscript.

Reviewer 2 Report

The authors proposed a review article, “Biomarkers Correlated with Tuberculosis Preventive Treatment 2 Response: a Systematic Review and Meta-analysis”. This review article is the first time to evaluate biomarker level change with TB preventive treatment. They have identified that INF-γ level change was related to TB preventive treatment. Minor corrections are needed before further processing.

1.     A funnel plot for bias with regression line and p-value should be presented.

2. The supplementary section should add detailed reasons for excluding studies from the analysis. And a summary of the included studies should be added in a table format.

3.     Table of test characteristics should be added, such as cut-off value.

Author Response

Response to Reviewer 2 Comments

The authors proposed a review article, “Biomarkers Correlated with Tuberculosis Preventive Treatment 2 Response: a Systematic Review and Meta-analysis”. This review article is the first time to evaluate biomarker level change with TB preventive treatment. They have identified that INF-γ level change was related to TB preventive treatment. Minor corrections are needed before further processing.

Point 1: A funnel plot for bias with regression line and p-value should be presented.

Response 1: As suggested, we provided the funnel plot in Supplementary Figure 1 and described it in lines 206-207 and lines 274-275.

Point 2: The supplementary section should add detailed reasons for excluding studies from the analysis. And a summary of the included studies should be added in a table format.

Response 2: As suggested, we added the detailed reasons for excluding studies from the analysis in Supplementary Table 4 and further specified it in line 216. Additionally, we added the included studies list in Supplementary Table 5 and the corresponding description in lines 222-223.

Point 3: Table of test characteristics should be added, such as cut-off value.

Response 3: We appreciated for your helpful suggestion. We added the test characteristics such as assay name and cut-off values in Table 1.

Round 2

Reviewer 1 Report

The authors addressed the comments.